# A Comparison between Crossbred (Holstein × Local Cattle) and Bangladeshi Local Cattle for Body and Milk Quality Traits

Sudeb Saha [1,2,*], Md. Nazmul Hasan [2], Md. Nazim Uddin [3], B. M. Masiur Rahman [4], Mohammad Mehedi Hasan Khan [5], Syed Sayeem Uddin Ahmed [6,*] and Haruki Kitazawa [1,7]

1 Food and Feed Immunology Group, Laboratory of Animal Food Function, Graduate School of Agricultural Science, Tohoku University, Sendai 980-8572, Japan; haruki.kitazawa.c7@tohoku.ac.jp

2 Department of Dairy Science, Faculty of Veterinary, Animal and Biomedical Sciences, Sylhet Agricultural University, Sylhet 3100, Bangladesh; liverausbargen@gmail.com

3 Department of Livestock Production and Management, Faculty of Veterinary, Animal and Biomedical Sciences, Sylhet Agricultural University, Sylhet 3100, Bangladesh; uddinmn.alm@sau.ac.bd

4 Bangladesh Food Safety Authority, Ministry of Food, Dhaka 1000, Bangladesh; bmmasiur@gmail.com

5 Department of Biochemistry and Chemistry, Faculty of Biotechnology and Genetic Engineering, Sylhet Agricultural University, Sylhet 3100, Bangladesh; mehedi2001bdbd@gmail.com

6 Department of Epidemiology and Public Health, Faculty of Veterinary, Animal and Biomedical Sciences, Sylhet Agricultural University, Sylhet 3100, Bangladesh

7 Livestock Immunology Unit, International Education and Research Center for Food and Agricultural Immunology (CFAI), Tohoku University, Sendai 980-8572, Japan

* Correspondence: sudeb.ds@sau.ac.bd (S.S.); ahmedssu.eph@sau.ac.bd (S.S.U.A.)

**Abstract:** Crossbreeding in dairy cattle with exotic breeds continues to be an appealing practice to the dairy farmers of Bangladesh. However, there is limited knowledge regarding the impact of crossbreeding on both the physical attributes and milk quality traits of crossbred cattle in Bangladesh. Therefore, the primary objective of this study was to evaluate the impact of crossbreeding Bangladeshi local cattle with the exotic Holstein breed on their body characteristics and milk quality. To achieve the goal, data pertaining to body traits and milk samples were gathered from a total of 981 cows from 19 dairy farms located in the northwestern region of Bangladesh. A trained evaluator measured body condition score (BCS), udder score, locomotion score, and body conformation traits. Milk yield information was acquired from official records, while milk composition details were determined through milk analysis. Notably, crossbred cows (Holstein × Local cattle) exhibited greater values for wither height (141 vs. 135, cm), body length (157 vs. 153, cm), heart girth (211 vs. 204, cm), BCS (3.69 vs. 3.27), and udder score (3.29 vs. 2.08) than their Bangladeshi local counterparts. Furthermore, crossbred cows produced 42.4% and 35.3% more milk (10.89 vs. 7.65, kg/d) and fat-corrected milk (10.35 vs. 7.54, kg/d) than Bangladeshi local cattle. However, milk from crossbred cows displayed lower fat and protein content, although their somatic cell score (SCS) and energy-corrected milk remained similar. Additionally, milk from crossbred cows exhibited a longer coagulation time when compared to that of Bangladeshi local cattle. In conclusion, crossbred cows (Holstein × Local cattle) had improved body characteristics with greater milk yield than Bangladeshi local cattle; however, lower fat and protein contents in milk with longer coagulation time were noted.

**Keywords:** crossbreeding; dairy cows; body conformation; milk production; milk composition

## 1. Introduction

Dairy farming remains an important subsector of the agricultural sector in Bangladesh, providing a vital source of income and nutrition for rural communities while contributing to the national economy [1]. As the demand for dairy products continues to rise, as a result, it must need to ensure sustainable production of dairy products. Proper breeding and genetic improvement can be a vital technology to improve dairy production. In this context, crossbreeding in dairy cattle has gained momentum in developed and developing countries

over the past few decades, primarily due to its positive effects on milk production, improved health, and fertility [2]. In developed nations like New Zealand, crossbreeding accounts for approximately 59.2% of the dairy cow population [3], while in Denmark and Sweden, crossbred cattle constitute 12% and 8% of the total cattle population, respectively [4]. Similarly, in developing countries such as India and Bangladesh, crossbred dairy cows comprise 26.5% and 8% of the overall cattle population, respectively [5,6].

Holstein stands out as the predominant dairy breed globally, owing to its remarkable capacity to produce high volumes of milk and its substantial body size [7]. Additionally, Holstein cows have the capacity to mobilize body reserves to achieve their genetic potential for milk production [8]. Consequently, Holstein has become a preferred choice for crossbreeding initiatives to enhance milk yield and improve cattle size for high carcass value. On the other hand, Bangladeshi local cattle (zebu cattle) are smaller in size and produce low amounts of milk. Crossbreeding a local breed (indigenous zebu type) with Holstein exhibited heterosis and additive effects for the milk yield and birth weight of calves [9]. Thus, interest in crossbreeding Holstein with other breeds and zebu cattle has surged, leading to numerous studies comparing the milk production and quality of crossbred cows with Holstein cows in diverse environmental conditions [2,6,10–13]. However, many of these studies have focused on crossbreeding Holstein cows with various recognized cattle breeds, often emphasizing milk production, quality, cheese yield, fertility, and survivability.

A staggering 92% of the dairy cattle population in Bangladesh comprises non-descriptive local cattle, characterized by low milk production and fertility rates [14]. Considering this situation, pure Holstein cows were imported with the aim of increasing national milk production, but their performance was considerably low in tropical environments like Bangladesh compared to temperate environments [15]. Additionally, this breed showed lack of heat tolerance and disease resistance capacity. Consequently, there is growing interest among Bangladeshi farmers in crossbreeding pure Holstein cows with local breeds to enhance milk production and improve cattle size for high carcass value. Culled cows also represent a vital source of meat production, and the economic value of culled cows depends on body size, conformation and weight, which directly affect the market price [16]. Given the importance of milk production and cattle size, it is deemed necessary to observe the field study evidence of this breeding strategy. To our knowledge, there is a shortage of reports on body conformation and milk composition traits for crossbred (Holstein × Local cattle) cows in Bangladesh. Most dairy producers in the region primarily focus on aspects such as body conformation, milk yield, and composition. Hence, the objective of this research was to compare crossbred (Holstein × Local cattle) cows with Bangladeshi local cattle in terms of their body characteristics and milk quality traits.

## 2. Materials and Methods

A total of 981 cows (from first to fifth lactation) were included in this study from 19 commercial dairy farms located in the northwestern region of Bangladesh. These farms have been implementing a crossbreeding program for several years, utilizing Holstein sire frozen semen imported from the US and locally produced semen from Holstein sires for the breeding process. Farms who were willing to participate and maintain farm records properly were considered for this study. This study was carried out from February to March 2022 for the collection of data and milk samples. The farms ranged in size from 30 to 82 cows with a mean farm size of 48 cows. In all herds, crossbred (Holstein × Local Cattle) and Bangladeshi local cattle were reared together. A typical representative of crossbred (Holstein × Local Cattle) and Bangladeshi local cattle in Figure 1. The dataset consisted of information collected from 384 Bangladeshi local cattle and 597 F1 crossbred (Holstein × Local cattle) cows. The average parity of crossbred (Holstein × Local Cattle) and Bangladeshi local cattle was 2.60 and 2.73, respectively. The average days in milk crossbred (Holstein × Local Cattle) and Bangladeshi local cattle was 175 and 169 days, respectively. The management practices across these herds were highly consistent, with all cows being nourished through a total mixed ration. This ration was meticulously prepared

by blending concentrate mixtures, green grass, and rice straw, ensuring that all experimental diets adhered to the recommendations outlined in the National Research Council (NRC) [17] guidelines 2001. Additionally, the cows were accommodated in a tie-stall barn and subjected to uniform management procedures throughout the entire duration of the experiment.

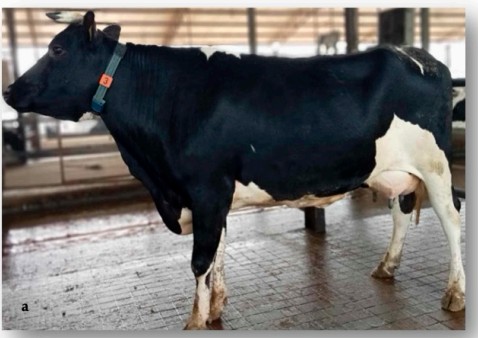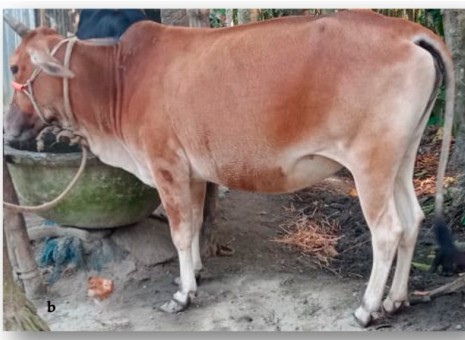

**Figure 1.** A typical representative of cattle: (**a**) crossbred (Holstein × Local Cattle) and (**b**) Bangladeshi local cattle.

The evaluation of cow body condition score (BCS) was conducted once by a trained evaluator on the same day of milk collection, following a scale ranging from 1 (indicating very thin) to 5 (indicating very fat), with increments of 0.25. This assessment method aligns with the technique established by Edmonson et al. in 1989 [18].

Udder score was noted by the same evaluator and recorded udder scores from 1 to 5 according to the guidelines described by Beard et al. (2019) [19]. The udder score combines udder conformation and a teat scoring system. An udder score of 1 or 2 consisted of pendulous udders and large teats, whereas 3 to 5 consisted of tight udders and small, symmetrical teats.

The evaluator recorded the locomotion score using a five-point scale (Manson and Leaver, 1988) [20], where 1 = no unevenness in gait or tenderness, and 5 = difficulty in walking and adverse effects on behavior pattern.

Additionally, measurements for wither height (WH) and body length (BL) were acquired once using a measuring stick. Wither height was determined from the floor to the highest point of the withers, while BL was measured from the scapular joint to the pin bone (as illustrated in Figure 2). Furthermore, heart girth (HG) measurements were taken using a tape measure positioned behind the front legs and shoulder blades.

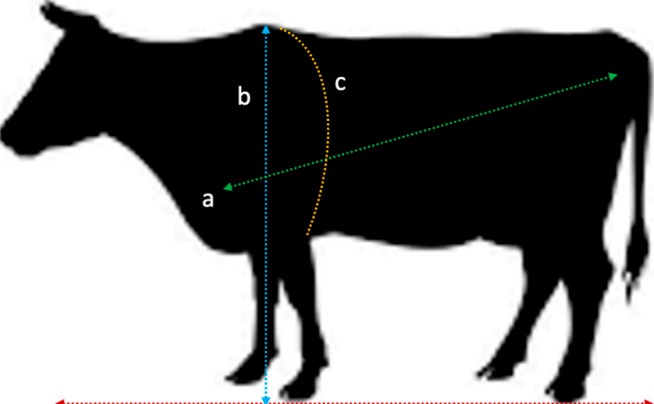

**Figure 2.** Scheme of body measurements for body length a, withers height b, and heart girth c of cow.

Milk yield data were retrieved from the official milk recording system. For composition analysis, milk samples (100 mL from each cow) were collected from all cows in 1 milk

recording sampling session per farm on the same day morning as the measurement of body conformation traits. These samples were promptly refrigerated at 4 °C and transported to the Department of Dairy Science, Sylhet Agricultural University laboratory. Within 12 h of collection, the samples were subjected to thorough analysis. Various milk components, including fat, protein, and lactose content, were meticulously analyzed using an ultrasonic milk analyzer (MT-25, Wincom company Ltd., Hunan, China), as outlined in detail by Saha et al. in their publication [21]. Milk component data were used to estimate the 4% fat-corrected milk (FCM) yield as 0.4 × milk yield (kg/d) + 15 × fat yield (kg/d) [22] Gaines 1928) and the energy-corrected milk (ECM) yield as (0.327 × milk yield) + (12.95% × fat × milk yield/100) + (7.65% × protein × milk yield/100) [23]. Milk pH was determined using a digital pH meter from Horiba Scientific (Kyoto, Japan). For the assessment of milk somatic cell count (SCC), a Eko-milk Scan (Somatic Cell Analyzer, Bulgaria) was employed, with subsequent transformation to Somatic Cell Score (SCS) facilitated by a log-transformation process as proposed by Ali and Shook in 1980 [24]. Furthermore, the measurement of milk coagulation time followed a technique described by Ikonen et al. in 2004 [25], with minor modifications. In brief, 5 mL of milk was poured into a teaspoon, to which 100 μL of clotting enzyme (rennet) was added. The aggregation process was then observed until its commencement.

*Statistical Analysis*

Before statistical analysis, cows were categorized according to parity (primiparous and multiparous) and days in milk (5 classes of 60 d each from 240 DIM, Supplementary Figure S1). Then, data were analyzed using a generalized linear model in Minitab 19 according to the following model:

$$y_{ijkl} = \mu + DIM_i + Parity_j + Breed_k + Herd_l + Breed_k \times Herd_l + e_{ijkl}$$

where $y_{ijkl}$ is each trait analyzed (milk, BCS, Udder Score, etc.); $DIM_i$ are the days in milk classes (i = 1, 2, 3, 4, 5), $Parity_j$ is the effect of parity (j = primiparous, multiparous), $Breed_k$ is the effect of the breed (k = Crossbred, Bangladeshi local cattle), $Herd_l$ is the effect of the herd (l = 1 to 19), $Breed_k \times Herd_l$ is the fixed effect of the interaction between the breed and the herd, and $e_{ijkl}$ is the experimental error term. All were included as fixed effects. Results are reported as least square means, and means were considered significantly different $p < 0.05$. Interaction between breed and herd was not significant and therefore not presented in the result. Additionally, interactions between parity, DIM, and herd data were not shown in this study.

## 3. Results and Discussion

The least squares means for body traits of Bangladeshi local cattle and F1 crossbred (Holstein × Local cattle) are summarized in Table 1. The average BCS was 3.5, as shown in Table 1. Notably, crossbred cows exhibited higher BCS values compared to the local cattle, aligning with the observations made by Mapiye et al. in 2010 [26]. Mapiye et al. (2010) [26] similarly reported that non-descript crossbreed cows displayed greater BCS than indigenous Nguni cattle. A possible explanation is that crossbred cows can utilize feed more efficiently than local purebred cows [27]. A study by Saha et al. (2018) [13] noted that three-way rotational crossbred cows had elevated BCSs in comparison to pure Holstein cows. It is important to note that Mushtaq et al. (2012) [28] observed a significant influence of breed on BCS, suggesting that crossbreeding may impact BCS due to the heterosis effect in cows [29]. Better BCS values seem to be closely related to the complementarity effect of the cattle used in the crossbreeding program. A study by Knob et al. (2021) [23] reported that crossbred cows exhibited higher BCS throughout lactation, which may lead to high productivity, health, and fertility.

**Table 1.** Least squares means and standard errors of BCS, locomotion score, udder score, and body conformation traits of Bangladeshi local cattle and F1 crossbred (Holstein × Local cattle) cows.

| Traits | Bangladeshi Local Cattle | Crossbred (Holstein × Local Cattle) | *p* Value |
|---|---|---|---|
| BCS [1] | 3.27 ± 0.03 | 3.69 ± 0.06 | <0.001 |
| Udder score | 2.08 ± 0.36 | 3.29 ± 0.28 | 0.05 |
| Locomotion score | 1.41 ± 0.35 | 1.33 ± 0.45 | 0.86 |
| Wither height, cm | 135 ± 0.59 | 141 ± 0.71 | <0.001 |
| Body length, cm | 153 ± 0.66 | 157 ± 0.87 | <0.001 |
| Heart girth, cm | 204 ± 1.09 | 211 ± 0.92 | <0.001 |

[1] BCS = Body condition score.

In this research, we observed that crossbred cows exhibited higher udder scores when compared to Bangladeshi local cattle. This finding aligns with the results reported by Pandit et al. in 2004 [30], which indicated that crossbred cows tend to have standard-to-high udder scores in comparison to native indigenous cattle. This difference may be attributed to the adoption of crossbreeding practices involving native indigenous cattle. It is worth noting that Holstein cows are recognized for their larger udders and are considered a superior dairy breed in terms of milk yield, as highlighted by Blöttner et al. in 2011 [10]. Consequently, crossing with Holstein cows may contribute to an improvement in udder size due to the additive effect of the Holstein gene. Interestingly, our study found that locomotion scores were comparable between local cattle and crossbred cows. This finding contrasts with the observations made by Singh et al. in 2018 [31], who noted that crossbred cows tend to be more susceptible to lameness, resulting in lower locomotion scores. It is important to consider that cow locomotion is closely related to the nutritional status of the cow and the conditions within their environment, as indicated by Oehm et al. in 2022 [32].

On average, the measurements of WH, HG, and BL for the sampled cows in our study closely resembled the values reported for Holstein cows and their crossbreeds, as documented by Hazel et al. in 2017 [33]. It is worth noting that Holstein and Ayrshire breeds are known for their larger body size in comparison to indigenous cows in Kenya, as highlighted by Lukuyu et al. in 2016 [34]. These larger-bodied animals possess a significant proportion of exotic genes, which, when introduced through crossbreeding, can enhance the body size of zebu cows. Our findings underscore the clear potential of crossbreeding to enhance the body size of local cattle. In fact, all body conformation traits, including WH, HG, and BL, were notably greater in crossbred cows when compared to their local counterparts.

The summary of least squares means for milk yield and quality traits between Bangladeshi local cattle and F1 crossbred (Holstein × Local cattle) is presented in Table 2. On average, the sampled cows produced milk at a rate of 8.7 kg per day, with fat and protein contents measuring 3.78% and 3.67%, respectively. In this study, crossbred cows (Holstein × Local cattle) demonstrated a remarkable 30% increase in milk yield compared to Bangladeshi local cattle. This finding aligns with the observations made by Garwe in 2001 [35], who reported that crossbred cows (Tuli × Jersey and Nkone × Jersey) yielded significantly higher volumes of milk than indigenous cows (Tuli and Nkone) in Zimbabwe. Similarly, other researchers have noted the positive impact of crossbreeding non-descriptive zebu cows with the semen of exotic dairy cattle on the milk productivity of these non-descript cows [5,36]. This increase in productivity may be attributed to the heterosis effects observed in crossbreeding studies for milk production in tropical countries. A study by Kunbhar et al. (2015) [37] reported that crossbred cows (Holstein × Red Sindhi) produced a greater amount of milk than Red Sindhi cattle. Holstein cows are known for high milk producing capacity; thus, crossbreeding with Holstein cows has been widely used to improve milk production in the tropics. In this study, milk from the crossbred cows exhibited lower fat and protein contents when compared to Bangladeshi local cattle, although the levels of lactose and SCS remained similar. This aligns with the findings of Islam et al. in 2008 [38], who reported that the milk of indigenous cattle had higher fat and protein contents than that of Holstein-crossed

indigenous cows. In the current study, crossbred cows (Holstein × Local cattle) had a 4% higher FCM than Bangladeshi local cattle, but ECM values were similar for both groups. High milk yield in crossbred cows could be the reason for the 4% higher FCM compared with Bangladeshi local cattle. On the other hand, Bangladeshi local cattle yielded milk with higher concentrations of fat and protein, as reflected by the ECM yield.

**Table 2.** Least squares means and standard error of milk yield and quality of Bangladeshi local cattle and F1 crossbred (Holstein × Local cattle) cows.

| Traits | Bangladeshi Local Cattle | Crossbred (Holstein × Local Cattle) | *p* Value |
|---|---|---|---|
| Milk yield, kg/d | 7.65 ± 0.82 | 10.89 ± 0.53 | <0.001 |
| Fat, % | 3.90 ± 0.12 | 3.67 ± 0.09 | 0.03 |
| Protein, % | 3.72 ± 0.09 | 3.64 ± 0.05 | 0.05 |
| Lactose, % | 5.06 ± 0.07 | 4.98 ± 0.06 | 0.62 |
| Milk pH | 6.48 ± 0.09 | 6.45 ± 0.06 | 0.06 |
| SCS [1] | 2.11 ± 0.28 | 2.35 ± 0.47 | 0.10 |
| FCM yield [2], kg/d | 7.54 ± 0.26 | 10.35 ± 0.31 | 0.002 |
| ECM yield [3], kg/d | 8.54 ± 0.15 | 11.77 ± 0.27 | 0.79 |
| Coagulation time, min | 10.83 ± 0.12 | 13.23 ± 0.91 | 0.05 |

[1] SCS = $3 + \log_2$ (SCC/100,000); [2] 4% fat-corrected milk yield (FCM): 0.4 × milk yield (kg/d) + 15 × fat yield (kg/d); [3] energy-corrected milk yield (ECM): (0.327 × milk yield) + (12.95% × fat × milk yield/100) + (7.65% × protein × milk yield/100).

Additionally, our results indicated that milk from Bangladeshi local cattle tended to have a higher pH than that from crossbred cows. Furthermore, milk coagulation time was significantly lower ($p < 0.05$) for Bangladeshi local cattle than crossbred cows. The coagulation time of milk is a crucial factor in cheese-making properties. Saha et al. in 2017 [2] found that milk from crossbred (Montéliarde × Holstein) cows had a shorter coagulation time than pure Holstein. Similarly, Teter et al. 2019 [39] reported that local Polish cattle exhibited a more favorable coagulation time than Holstein–Friesian cattle. In general, milk from Holstein (HO) cows tends to have a longer coagulation time [12,40], possibly due to the 50% Holstein blood in crossbred cows.

## 4. Conclusions

During the past decade, crossbreeding in dairy cattle has gained popularity in the tropical countries like Bangladesh for improving the milk production. In this study, we focused on a crossbreeding approach by mating Bangladeshi local cattle with Holstein sires. The results revealed that the resulting crossbred (Holstein × Local cattle) cows exhibited larger body sizes and higher body condition scores. Additionally, they produced significantly higher volumes of milk and 4% fat-corrected milk. However, it is worth noting that the milk from crossbred cows contained lower fat and protein levels, although lactose, energy-corrected milk, and somatic cell count remained similar. Despite the extended coagulation time observed in the milk from crossbred cows, there is potential for higher cheese yield due to their increased milk production compared to Bangladeshi local cattle. To provide a comprehensive comparison of the crossbreeding scheme with pure Holstein cows, further studies are necessary. These studies should investigate effects on cheese yield traits, health, fertility, and longevity.

**Supplementary Materials:** The following supporting information can be downloaded at: https://www.mdpi.com/article/10.3390/dairy5010012/s1, Figure S1: Number of animals belongs to crossbred (Holstein × Local Cattle) and Bangladeshi Local Cattle in different classes of days in milk (DIM).

**Author Contributions:** Conceptualization, S.S. and H.K.; methodology, M.N.H. and S.S.; software, S.S.U.A., M.M.H.K. and S.S.; validation, S.S., B.M.M.R. and H.K.; formal analysis, M.M.H.K. and M.N.U.; investigation, S.S. and H.K.; resources, S.S.; data curation, M.N.H. and M.N.U.; writing—original draft preparation, S.S.; writing—review and editing, S.S.U.A., H.K. and M.M.H.K.;

visualization, H.K.; supervision, S.S. and S.S.U.A.; project administration, S.S.; funding acquisition, S.S. and H.K. All authors have read and agreed to the published version of the manuscript.

**Funding:** This research was supported by Grant-in-Aid for JSPS Fellow (22F22080) from the Japan Society for the promotion of Science (JSPS) to H.K.

**Institutional Review Board Statement:** Not Applicable.

**Informed Consent Statement:** Not Applicable.

**Data Availability Statement:** Data is contained within the article or supplementary material.

**Acknowledgments:** The authors are thankful to the owners and managers of the herds for their cooperation and support during data and sample collection.

**Conflicts of Interest:** The authors declare no conflicts of interest.

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
