# Peer review of "A Comparison between Crossbred (Holstein × Local Cattle) and Bangladeshi Local Cattle for Body and Milk Quality Traits"

_2624-862X, doi:10.3390/dairy5010012_

Round 1
Reviewer 1 Report (Previous Reviewer 2)
Comments and Suggestions for Authors
The authors have addressed all of my concerns。
Author Response
Dear Reviewer,
We appreciate your precious time to review our manuscript. Please see the attached file.
Thank you!!
Best regards
Sudeb Saha, PhD

Reviewer 2 Report (New Reviewer)
Comments and Suggestions for Authors
This study aimed to compare the physical attributes and milk composition between Bangladeshi local cattle and crossbred (Holstein × Local cattle). The idea of this research or this practice is good and worth trying.
Specific comments:
- Is there a well-known name for the Bangladeshi local breed?
Lines 28-29: please delete, no need to mention the average for all samples and animals
Line 32: in Table 2, the value of milk is 7.65 kg while in this line it's 9.3 kg, in addition, it's not clear for which group this value is. Oh, I got it, this is the average value of milk production and composition for the two groups, please delete it.
Line 92: Please mention the average age and body weight for each group of cattle.
Lines 101-120: When were these measurements taken and how many times were they measured during the research study?
Line 107: don’t start any sentence with an abbreviation
Line 124: how many times during the study?
Lines 155-160: if the average of BCS for the crossbred cattle is 3.69, don’t you think it’s a little bit high for dairy cattle in all phases of milk production?
Overall, the available results are enough for a Communication type article.
Author Response
Dear reviewer,
Thank you for your time to review our manuscript. It was valuable and insightful comments that led to improves our manuscript. Please see the attached file for the responses of your comments.
Best regards
Sudeb Saha, PhD

Reviewer 3 Report (New Reviewer)
Comments and Suggestions for Authors
The manuscript entitled "A Comparison Between Crossbred (Holstein × Local Cattle) and Bangladeshi Local Cattle for Body and Milk Quality Traits" brings important findings to farmers from Bangladesh. However, several inconsistencies must be addressed before further consideration.
Abstract
Line 28: Whose body traits?
Lines 28-36: Please, rewrite the results of the Abstract section. It is a little bit confusing. Be assertive.
Liens 36-39: on the abstract only include a conclusion of your findings. The recommendation makes sense; however, it is not welcomed at this moment.
Line 40: All these words are already in the title. Please, replace them.
Introduction
Lines 44-46: Did the authors assess the conditions and management of dairy cows? Please, revise this sentence.
LIne 46-47: Please revise this sentence. This reviewer did not understand its contribution to the manuscript.
Lines 58-62: Please improve this sentence. It is not clear why is important to crossbreed a local breed with Holstein. Be assertive.
Lines 59: Why is important to increase cattle size?
Line 64: like?
Line 72: Again, why is it important to increase cattle size? Are bigger cows more efficient than smaller cows? The expression cattle size is not ok for this reviewer. Please revise.
Lines 72-76: Now, this reviewer understands why cattle size is important in this context. However, the previous ”cattle size” needs to be contextualized.
Materials and Methods
Line 91: crossbred (Holstein x Local cattle)
Line 107: Do not start a sentence with acronymous.
Lines 111-112: This sentence is not M&M. Please, remove it.
Line 125: Please, describe the procedure of collection.
Line 131: Please, remove “2019”.
Line 150: Why DIM and herd were considered a fixed effect? Parity interaction with breed should be provided.
Line: Please, revise the paragraphs of M&M aligning to the results appearance.
Results and Discussion
Lines 155-169: Please, be careful interpreting data from pure Holstein to crossbred Holstein with the findings of the present study. Generally, pure Holstein cows have lower BCS than crossbred dairy cows. So, crossbreeding Holstein cows with a local breed may increase BCS compared to pure Holstein. In this present study, the authors did not compare pure Holstein with crossbred Holstein cows. Please, revise the explanation herein given. Also, provide a table describing the average DIM, Parity proportion in each breed, and the proportion of local breeds and crossbred cows by herd size. Does bigger herds higher proportion of crossbred cows compared to smaller herds?
Lines 167-169: which type of crossbred cattle? Please, describe it.
Line 177-178: this reviewer did not understand this explanation. Since the udder size of Holstein is bigger than local breed cattle, why udder pendulosity was reduced in the local breed, for example?
Line 201-204: Why this information is important?
Line 210-216: Please, revise this sentence adding some information about the Holstein selection for milk yield.
Table 2: Please, provide the analysis of fat-correct milk yield and energy-corrected milk yield. They are important outcomes to be analyzed.
Conclusion
Lines 231-236: Considering revise this sentence since there are not efficiency outcomes analyzed in this study. The input (nutrient intake) is important when talking about different breeds.
Line 238-240: Greater milk production did not mean greater efficiency. FCM and ECM analysis will improve the discussion of this manuscript.
Author Response
Dear Reviewer,
Thank you for your valuable comments and suggestions that led to possible improvements in the current version of manuscript. The authors carefully considered the comments and tried our best to address every one of them. Please see the attached file for the responses of your comments.
Again thanks for your time!
Best regards
Sudeb Saha, PhD

Round 2
Reviewer 3 Report (New Reviewer)
Comments and Suggestions for Authors
Thank you for your consideration of my comments and questions in round 1. The authors have made great efforts to address these. I have included two more suggestions below. These are listed by line number.
Abstract
Line 31: replace “Produce” by “Produced”. In addition, improve the sentence such as “…, crossbred cows produced 42.4% and XX% more milk (10.89 vs 7.65 kg/d) and fat corrected milk (XX vs XX kg/d) than Bangladeshi local cattle”.
Line 59: Improve the sentence. Rewrite it such as “Crossbreeding local cattle (describe the cattle breed) with Holstein…”
Author Response
Dear reviewer,
We appreciate you for your precious time in reviewing our paper. Please see the attached file for responses to your comments.
Thank you!!
Best regards
Sudeb saha, PhD

This manuscript is a resubmission of an earlier submission. The following is a list of the peer review reports and author responses from that submission.
Round 1
Reviewer 1 Report
Comments and Suggestions for Authors
This study compares performances of Holstein-crosses with local Bangladeshi cattle and provides useful knowledge for dairy produces in Bangladesh. The manuscript reads well and the statistical method for analyzing data seems appropriate.
Crossbreeding with exotic breeds is no doubt a good solution to fast and sustainable improvement of milk production in developing countries. However, I miss some discussion on the consequences of it. You seem to focus a lot on getting bigger cows to improve milk yield. Did you consider the downside of having bigger cows in barns made for smaller cows? Also, bigger cows eat more and so do higher yielding cows. Since you did not measure feed intake, you cannot estimate feed intake. But I think you ought to discuss the nutritional requirement of larger and higher yielding cows compared to the actual gain in milk yield. If the crossbred cows are kept in suboptimal conditions, they may not be more economically feasible than the local breeds. In relation to this, I would also like you to briefly discuss whether it is more feasible to do crossbreeding versus having purebred Holstein in Bangladesh.
Lines 29-32 (abstract): The results do not match the numbers in tables 1 and 2.
Lines 32-33 (abstract): Write out how much more milk they produced in % or kg.
Introduction: Since you compare conformation characteristics in your study, I think you should write a few sentences on how body composition relates to production.
Lines 73-77: For how long were these cows observed? And which lactation numbers? Also, I would like to know more about the Holstein-genetics used for crossbreeding. Was the semen imported from the US, Europe or somewhere else? Or was the semen derived from local Holstein populations in Bangladesh?
Lines 77-79: Were the local cattle and the crossbreds within the same herds or in different herds? If they were in different herds, your results may be biased by different management practices (not just feeding) related to breed.
Lines 93-95: Why was it relevant to record locomotion? I would assume lameness, but it is not clear at this point in the paper.
Lines 142-147: I don’t think it’s the heterosis effect that improves the udder conformation in the crossbreds, but simply the additive effect of Holstein genes. As you mention, the Holstein breed is known for its superior udder conformation. Hence, the complementarity of crossing Holstein with local cattle will highly improve the udders of the crossbred cows.
Lines 147-152: I think the reason why you see more lame crossbred cows is because they are also somewhat bigger than the local cattle. My assumption is, that they are put in stalls made for local cattle, i.e., stalls that may be too small for the crosses. However, your results on locomotion scores were insignificant, so my assumption is vague.
Lines 158-167: Why is it important to increase the size of the cow? I’d like to challenge you a bit here. True, a larger frame of the cow leads to a potential for a bigger udder. But you could also get a larger cow by crossbreeding with a large exotic beef breed (for example Limousin or Charolais) – but that doesn’t lead to higher milk yield, because these breeds are not selected for that. I dare to guarantee you that you would get great crossbreeding results as well from crossbreeding with Jersey, even though it is a smaller cow (you have a reference on that later in line 174).
Author Response
Dear Reviewer,
We appreciate your precise time for reviewing our manuscript. Please see the attached file for the response to you.
Best Regards
Sudeb Saha, PhD

Reviewer 2 Report
Comments and Suggestions for Authors
Manuscript ID:dairy-2624470
Titile: A Comparison Between Crossbred (Holstein × Local Cattle) and Bangladeshi Local Cattle for Body and Milk Quality Traits
This manuscript investigated the impact of crossbreeding Bangladeshi local cattle with the exotic Holstein breed on their body conformation and milk traits. The results suggest that crossbred cows exhibited greater values for WH, BL, HG, BCS, and udder score than Bangladeshi.Crossbred cows produced a higher volume of milk compared to Bangladeshi local cattle. Milk from crossbred cows displayed lower fat and protein content. These results are useful for dairy industry. I have several minor comments.
I suggest to put photo of crossbred cows and Bangladeshi local cattle in Figure 1.
In Statistical Analysis section. It is better to regard 19 commercial dairy farms as fix effects.
Author Response
Dear Reviewer,
Thank you for reviewing our manuscript and providing your valuable comments.
Best regards
Sudeb Saha, PhD

Reviewer 3 Report
Comments and Suggestions for Authors
The authors compared crossbred (Holstein × Local cattle) cows with 70 Bangladeshi local cattle in terms of their body characteristics and milk quality traits. However, there is no sound justification and rationale for conducting this study and I can't find the pertinent findings that can be extrapolated outside Bangladesh dairy industry. Hence, this article will not benefit from international readership. The authors may consider publishing the results in a local or regional journal.
Materials and Methods
The methodology has several flaws. There was no statement regarding the study design, sampling method, sample size calculation and inclusion criteria for the farms selected. The authors merely jumped into stating the number of cows recruited from 19 commercial farms.
In addition, no detailed information was provided as per the trained assessors in terms of intra- or inter assessor agreement. Data analysis was briefly presented without any information on criteria used in selecting the statistical tests employed.
I won't comment on the results and discussion session for given the extensive limitations in the research objectives and methodology.
Comments on the Quality of English LanguageModerate revision of English language is required
Author Response
Dear Reviewer,
Thank you. Please find the attachment for reviewer comments and reply.
Best regards
Sudeb saha, PhD
